

# Novel glauconite compounds improve soil properties and sugar beet (*Beta vulgaris* L.) yields in saline soils

Mahmoud El-Sharkawy[1], Modhi O. Alotaibi[2,3], Esawy Mahmoud[1], Kholoud A. El-Naqma[4], Ramadan E. Kanany[4], Medhat G. Zoghdan[4] and Mahmoud M. Shabana[4]

[1] Soil and Water Department, Tanta University, Tanta, Egypt

[2] Department of Biology, College of Science, Princess Nourah bint Abdulrahman University, Riyadh, Saudi Arabia

[3] Environmental and Biomaterial Unit, Natural and Health Sciences Research Center, Princess Nourah bint Abdulrahman University, Riyadh, Saudi Arabia

[4] Soils, Water and Environment Research Institute (SWERI), Agricultural Research Centre, Giza, Egypt

Corresponding authors
Mahmoud El-Sharkawy,
mahmoud.elsharkawy@agr.tanta.edu.eg
Modhi O. Alotaibi,
mouotaebe@pnu.edu.sa

## ABSTRACT

Sugar beet is essential for sugar production, supporting food industries and renewable energy resources. A two-season field experiment (2021/2022 and 2022/2023) evaluated the effects of different potassium (K) sources, including traditional potassium sulfate (K), glauconite powder (G), and foliar glauconite extracts (20- and 40-mL L$^{-1}$) extracted with sulfuric acid (GS), humic acid (GH), or hot water (GW), on soil properties, sugar beet yield, and sugar quality in saline soils. The results showed that GH and G treatments significantly improved soil properties by reducing electrical conductivity (EC), exchangeable sodium percentage (ESP), and bulk density (BD), while increasing organic matter (OM) and infiltration rate (IR). The application of glauconite extracted by humic acid in 40 mL (GH2) treatment improved soil nutrient availability, notably increasing nitrogen (by 73.4%), potassium (by 137.2%), cupper (by 219.7%), and manganese (by 316.7%) compared to control, while GS2 enhanced ferrous (by 213.7%) and zinc (by 363.7%). GH2 application led to remarkable improvements in sugar beet yield, with average increases in root yield (94.84%), shoot yield (100.45%), total sugar yield (137.22%), and sucrose (20.5%) compared to the control, whereas GW treatments showed the least improvements. Despite GW recording the lowest sugar impurities, GH2 recorded the lowest alkalinity level. Advanced analytical techniques such as heatmaps, self-organizing maps (SOM), and while non-metric multidimensional scaling (NMDS) analysis revealed strong positive correlations between soil properties, sugar beet responses and sugar quality attributes. These findings underscore the potential incorporation of new innovative cost-effective alternative foliar amendment derived from glauconite rock-waste extraction with humic acid highlighting a sustainable agricultural strategy for managing soil health and crop productivity contributing to food security and environmental sustainability.

## INTRODUCTION

Sugar beet (*Beta vulgaris* L.) is a second major global crop for sugar production, contributing significantly to the world's sugar supply alongside sugarcane. It serves as a major source of sucrose, accounting for approximately 20% of the world's sugar supply, with sugarcane contributing the remainder (*Hoffmann, Koch & Märländer, 2021*). As global demand for sugar rises, sugar beet cultivation has expanded into diverse and challenging environments, including arid and semi-arid regions where saline soils hinder crop productivity (*Tayyab et al., 2023*; *Wang et al., 2024*). Egypt, where agriculture is constrained by limited arable land and increasing soil salinity and water scarcity, has considered sugar beet as a valuable crop for both sugar production and crop diversification (*El-Mouhamady, Al-Kordy & Elewa, 2021*). The Egyptian government encourages sugar beet cultivation to reduce dependence on sugar imports, which helps strengthen food security and supports local sugar industries. Sugar beet has become one of Egypt's most valuable crops, particularly in the Nile Delta region, where it is cultivated on over 600,000 feddans (approximately 252,000 hectares), contributing significantly to the country's sugar production (*El-Zayat, 2021*).

In Egypt, saline soils are widespread, particularly in the northern Nile Delta, where irrigation and intensive farming practices contribute to high levels of soluble salts and exchangeable sodium (*Aboelsoud et al., 2022*). These conditions result from a combination of natural soil properties, irrigation practices, improper fertilization managements, and climate factors. Potassium (K) is an essential nutrient that enhances plant tolerance to salinity, improving water-use efficiency and physiological processes like photosynthesis and sugar accumulation (*Mostofa et al., 2022*). In addition to conventional potassium fertilizers, alternative sources such as glauconite "a potassium-bearing mineral abundant in Egypt" have gained interest for their potential to provide a slow-release K source that may enhance soil properties and plant growth under adverse conditions (*Oze et al., 2019*).

Glauconite, locally sourced in the Western Desert of Egypt, presents a promising alternative as a potassium fertilizer. Its content of potassium, iron, and other micronutrients can help alleviate the nutrient limitations of saline soils by gradually releasing K and other elements (*Dasi, Rudmin & Banerjee, 2024*). However, the nutrient availability in the solid form of glauconite may be limited due to its low solubility. To increase its effectiveness as a K-source, we hypothesize that various extraction techniques to release potassium from glauconite could enhance its impact on plant growth. Extraction methods involving sulfuric acid, humic acid, and hot water have been studied for other minerals in laboratory scales but have not been extensively investigated for glauconite in the context of Egyptian agriculture. Sulfuric acid extraction has been extensively studied for releasing potassium from silicate minerals, demonstrating high efficiency in increasing K availability (*Krupskaya et al., 2015*). Humic acid, a natural organic chelator and chemical reactive agent, has shown potential in improving nutrient solubility and availability, especially in saline and sodic soils, making it a promising eco-friendly alternative (*Dasi, Rudmin & Banerjee, 2024*). Hot water extraction, though simpler and cost-effective, is hypothesized to release water-soluble potassium fractions while preserving the mineral's structural integrity. The application of extracted glauconite using these methods represents a novel approach to enhance its

efficacy as a potassium source in saline soils, and its impact on sugar beet yield and quality has yet to be investigated. This study aims to investigate the effects of various potassium sources, including conventional potassium sulfate, raw glauconite, and different glauconite extracts (sulfuric acid, humic acid, and hot water), on soil fertility, sugar beet yield, and sugar quality under saline conditions. Despite glauconite's abundance as a potassium-rich silicate mineral, its agronomic application has been limited by the slow release of potassium in its raw form. Novel extraction techniques using sulfuric acid, humic acid, and hot water have been explored to enhance glauconite's potassium availability. However, a comprehensive evaluation of these methods under field conditions, particularly in saline-affected soils, remains scarce. This research bridges this gap by comparing these innovative extraction techniques and assessing their impact on soil health, nutrient dynamics, and sugar beet productivity. Furthermore, it aims to identify the most effective and sustainable glauconite extraction method, offering a cost-effective and environmentally friendly alternative for improving crop productivity and managing saline soils in Egypt and similar regions.

## MATERIALS & METHODS

### Experimental layout and location

A two seasons field experiment was conducted in North Nile Delta (Al-Hamul District, Kafr El-Sheikh Governorate, Egypt), during the successive winter seasons of 2021/2022 and 2022/2023. The study aimed to investigate the influence of different potassium sources and application methods on saline soil properties irrigated with poor-quality water and their impact on sugar beet (*Beta vulgaris* L.) productivity. Potassium was applied either as a soil addition (potassium sulfate and glauconite powder) or through foliar application using various glauconite extracts. The experiment was carried out at Shabana's field located at 31°25′29″N Latitude, 31°04′23″E Longitude with an altitude of six m above sea level. The experimental was approved and supervision provided by Soils, Water and Environment Research Institute (SWERI). The classification of the soil is "vertisols" and the soil characteristics before planting are presented in Table 1. The field was divided into 36 plots, each measuring 15 m in length and 5 m in width. The experiment followed a completely randomized design with four replicates. The treatments included: control without K-addition (CK)- potassium sulfate at the recommended rate of K of 24 kg K fed$^{-1}$(K), glauconite powder in 100% K (G), foliar application of 20 ml L$^{-1}$ glauconite extracted with sulfuric acid (GS1), Foliar application of 40 ml L$^{-1}$ glauconite extracted with sulfuric acid (GS2), foliar application of 20 ml L$^{-1}$ glauconite extracted with hot water (GW1), foliar application of 40 ml L$^{-1}$ glauconite extracted with hot water (GW2), foliar application of 20 ml L$^{-1}$ glauconite extracted with humic acid (GH1), and foliar application of 40 ml L$^{-1}$ glauconite extracted with humic acid (GH2).

Soil preparation included plowing and subsoiling at 1.5 m spacing and 45 cm depth across all treatments. The field was leveled using LASER technology, achieving a surface slope of 0.1%, with long graded furrows (60 m in length).

Sugar beet seeds (*Beta vulgaris* L. *var pleno*) were sown at a rate of 4 kg fed$^{-1}$ after maize in the first week of October during both seasons. Plants were thinned at the 4-leaf

**Table 1** Mean values for physical and chemical properties of the experimental soil (0–30 cm) before cultivation in 2023 and 2024 seasons.

| Traits | Values | 1st Season | | 2nd Season | |
|---|---|---|---|---|---|
| | | Soil | Water | Soil | Water |
| pH | – | 8.64[$] | 8.06 | 8.65[$] | 8.12 |
| EC | dS m$^{-1}$ | 4.21[$] | 0.67 | 3.85[$] | 0.65 |
| SAR | % | 12.1 | 3.97 | 11.22 | 3.91 |
| ESP | % | 14.41 | – | 13.51 | – |
| Available (N) | mg kg$^{-1}$ | 42.31 | 3.94[*] | 45.43 | 3.83[*] |
| Available (P) | mg kg$^{-1}$ | 6.56 | – | 5.87 | – |
| Available (K) | mg kg$^{-1}$ | 178 | – | 212 | – |
| OM | % | 1.16 | – | 1.23 | – |
| CaCO$_3$ | % | 2.31 | – | 2.24 | – |
| Texture class | – | Clayey | – | Clayey | – |
| FC | % | 43.42 | – | 45.24 | – |
| WP | % | 23.42 | – | 24.18 | – |
| Bulk density | kg m$^{-3}$ | 1.45 | – | 1.44 | – |
| Total porosity | % | 45.28 | – | 45.66 | – |
| PR | N cm$^{-2}$ | 380 | – | 370 | – |
| IR | cm hr$^{-1}$ | 0.72 | – | 0.73 | – |
| Total Mn | mg kg$^{-1}$ | 22.21 | 1.25 | 23.14 | 1.24 |
| Total Fe | mg kg$^{-1}$ | 3.5 | 3.27 | 3.3 | 3.23 |
| Total Zn | mg kg$^{-1}$ | 0.23 | 0.13 | 0.24 | 0.12 |
| Total Cu | mg kg$^{-1}$ | 0.57 | 0.014 | 0.61 | 0.016 |

**Notes.**
[*]total nitrogen.
[$]soil paste extraction.
EC, electrical conductivity; SAR, sodium adsorption ratio; ESP, exchangeable sodium percentage; OM, organic matter; PR, penetration resistance; WP, water welting point; FC, field capacity; IR, hydraulic conductivity.

stage to one per hill before the first irrigation. Each plot received 200 kg fed$^{-1}$ of calcium superphosphate (15.5% $P_2O_5$) during soil preparation. Nitrogen fertilizer (90 kg N fed$^{-1}$) as urea (46% N) was applied in two doses: the first after thinning, and the second 30 days later. Potassium sulfate (48% $K_2O$) was applied at the rate of 50 kg fed$^{-1}$ only in the K-treatments plots in one dose with the firs irrigation. The solid glauconite was dried, ground, and applied at 480 kg fed$^{-1}$ during soil preparation, corresponding to the recommended potassium rate for sugar beet. The chemical composition of the powder glauconite is as follows: $K_2O$ (5.40%), $Fe_2O_3$ (15.17%), CaO (4.23%), MnO (0.05%), $Al_2O_3$ (1.04%), $SiO_2$ (52.26%), $TiO_2$ (0.51%), MgO (1.17%), $P_2O_5$ (0.24%), $Na_2O$ (0.75%), and loss on ignition (LOI) is (9.18%). All other pursuits are controlled according to the guidelines supplied by the Ministry of Agriculture recommendation for sugar beet plants in the North Delta area. Sugar beet plants were harvested after 180 days of planting on 27th of April 2022 first season, and on 24th of April 2023 for the second season.

The extraction method for glauconite was carried out as follows: 200 g of dry glauconite powder was placed into one L flasks, and the volume was filled with the respective extraction solutions, which included $H_2SO_4$ (2N), pure humic acid (99.9%), and hot deionized water

**Table 2 Extractable glauconite properties.**

| Treatments | pH | EC | N | P | K | Na | Cu | Fe | Mn | Zn |
|---|---|---|---|---|---|---|---|---|---|---|
| | – | dS m$^{-1}$ | % | % | Meq L$^{-1}$ | Meq L$^{-1}$ | ppm | ppm | ppm | ppm |
| GS | 1.01 | 26.10 | 0.756 | 0.013 | 52.2 | 150 | 1.74 | 28,951.25 | 501.80 | 11.50 |
| GW | 4.5 | 4.95 | 0.350 | 0.004 | 4.8 | 30 | ND | ND | ND | ND |
| GH | 1.61 | 22.20 | 0.406 | 0.022 | 95 | 90 | 6.04 | 9,060.00 | 717.30 | 44.65 |

Notes.
GS, Glauconite extracted with sulfuric acid; GW, Glauconite extracted with hot water; GH, Glauconite extracted with humic acid; ND, not determined.

(100 °C). The mixtures were shaken for 24 h at 4,000 rpm at room temperature. After shaking, the supernatants were filtered and stored in dark bottles until use. The chemical composition of each extract is shown in Table 2.

## Soil characteristics

Surface soil samples were gathered at a depth of 30 cm from each experimental plot after harvesting. The samples were air-dried, crushed, sifted to pass through a 2.0 mm sieve then homogenized. Soil pH (1:2.5; w:v) and electrical conductivity (1:5; w:v) were determined using a standard pH-meter (model H12211-02, Thermofisher, HANNA, USA), and conductivity meter (model CON2700, EUTECH, USA), respectively (*Pansu & Gautheyrou, 2006*). Total organic matter was measured by wet digestion using $K_2Cr_2O_7$ (*Sleutel et al., 2007*), while cation exchange capacity (CEC) was determined using $CH_3CO_2NH_4$ as outlined by *Cottenie, Verloo & Kiekens (1982)*. The concentrations of available P and K in soil were determined according to methods reported by *Page, Miller & Keeney (1983)*, while available N ($NH_4$ an $NO_3$) was determined using the Kjeldahl method (*Varley, 1972*). Soil bulk density (BD) and total porosity (TP) were assessed according to *Campbell & Henshall (2000)*. A hand penetrometer device was used to determine the soil penetration resistance (SPR) as reported by *Herrick & Jones (2002)*. Particle size distribution was determined by the pipette method of *Scheldrick (1993)* and calcium carbonate was measured volumetrically using calcimeter (*Şenlikci et al., 2015*). SPR was determined by hand penetrometer device (*Herrick & Jones, 2002*). The water infiltration rate in soil were determined according to *Brar et al. (2015)*. Total concentrations of heavy metals including Mn, Fe, Zn, and Cu were determined after digestion by concentrated $H_2SO_4+H_2O_2$ using ICP Spectroscopy (ICP-ISO Prodigy Plus) as reported by *Page, Miller & Keeney (1983)*. Soil sodicity indice was assessed by calculating the exchangeable sodium percentage (ESP) based on the method described by *El-Sharkawy, El-Aziz & Khalifa (2021)*.

## Plant biomass and chemical analysis

Ten plants from each treatment were selected randomly at the maturity stage to measure plant biomass and chemical properties. The productivity parameters including shoot yield (t Fed$^{-1}$) and root yield (t Fed$^{-1}$) were determined as outlined by *Nassar et al. (2023)*, while sugar yield (kg fed$^{-1}$) was determined by multiplying root yield with sugar percentage as assessed by *Koch et al. (2018)*. A hand refractometer (model BK-PR, Biobase, Shandong, China) was used to record the extracted sugar (%) in the fresh root juice as reported by *Carruthers & Oldfield JFTBT-TTV of the SB (2013)*.

## Sugar quality properties

In the root fresh juice, the alkalinity of the sugar was determined using pH-meter, while the percentage of sucrose was measured according to the method reported by *Le Docte (1927)*. Sodium and potassium impurities were determined in the digested solution followed the method of *Havre (1961)* using Flame-photometer (model PFP7/C, Jenway, NJ, USA), which is a well-established analytical tool for accurately quantifying alkali metals in liquid samples. while $\alpha$-amino N was determined using the hydrogenation protocol as described by *Cooke & Scott (2012)*. Regarding the equation of *Kenter & Hoffmann (2009)*, the sugar losses to molasses percentage (SLM%) was computed as the follows:

$$\text{SLM}(\%) = 0.14(\text{Na} + \text{K}) + 0.25(\alpha - \text{amino N}) + 0.5. \tag{1}$$

Sugar purity index was calculated as described by *Smith, Martin & Ash (1977)* as the following:

$$\text{Purity }(\%) = \frac{\text{Extracted sugar}(\%)}{\text{Total Soluble solids}(\%)} \times 100. \tag{2}$$

## Statistical analysis

The experiments were analyzed using one-way analysis of variance (ANOVA) with four replicates by IBM-SPSS statistics (version 29, IBM Corp., Armonk, NY, USA). Replications were considered random, and all other variables were treated as fixed effects with significant levels set to 5%. A Duncan multiple range test (DMRT) was conducted to compare the means at significance ($p < 0.05$). The multivariate statistics using principal component analysis (PCA), non-metric multidimensional scaling (NMDS) analysis, and self-organizing maps (SOM) have been carried out using MATLAB software (v. R2022a, The MathWorks, Natick, MA, USA).

# RESULTS

## Effect of glauconite amendments on soil properties

The data in Table 3 illustrated that different sources of K and glauconite extracts did not affect significantly in soil pH in both seasons. As for OM, the application of glauconite resulted in double the values compared to the control followed by the foliar application of 40 mL L$^{-1}$ glauconite extract with humic acid (GH2) recording 1.94% and 1.97% for the first and second seasons respectively. Different treatments affected significantly ($p < 0.01$) in soil salinity through EC, and sodicity through ESP. The data showed that the foliar application of glauconite extract with humic acid particularly CH2 decreased the EC reaching a reduction percentage of 30% compared to control in both seasons followed by G treatment with no significant difference. Additionally, the ESP takes the same pattern in alleviating sodicity stress by the application of GH2 with a reduction of 24.22% and 16.03% for the first and second seasons respectively followed by GW1 (20.8%) in an average of both seasons compared to control. Regarding the physical properties, the treatments affected significantly ($p < 0.01$) in both IR and SPR, while it had no effects on BD. The

foliar application with GH in both concentrations registered the highest water infiltration rate in the soil with no differences with (G) treatment in both seasons recording the highest values with GH2 of 1.17 and 1.18 cm hr$^{-1}$ in the first and second seasons, respectively. The soil penetration resistance (SPR) exhibited fluctuated changes in both seasons recording the lowest record in the first season with (G) treatments (250 N cm$^{-2}$) followed by (K) treatments (280 N cm$^{-2}$), while in the second season recorded 240 N cm$^{-2}$ and 270 N cm$^{-2}$ for G and K treatments, respectively.

The soil nutrient compositions are presented in Table 4. The foliar application with GS2 resulted in increasing NH$_4$ in the first season with no difference with GH2, while the NO$_3$ followed a different trend, with glauconite (G) treatment recording the highest rate. On the other hand, GH2, and GH1 registered the highest rates of NH$_4$ and NO$_3$ in the second season with increments 2 and 3 times compared to control treatments respectively. Despite these results, the total—N was registered at the highest levels with the application of G recording 87.99 and 92.88 mg kg$^{-1}$ in the first and second seasons respectively. The GH2 showed a magnificent increase in K compared to other treatments in both seasons. Specifically, it recorded 2.6 times higher K content than the control in the first season and 2.2 times higher in the second season. In contrast, GW1 recorded the least enhancement level of K with an average increment of only 2.71% across both seasons compared to the control. The micronutrients including Fe, Mn, Zn, and Cu influenced significantly ($p < 0.01$) with different treatments. The data showed that GS2 increased the soil content of Fe and Zn recording 12.09 and 0.82 mg kg$^{-1}$ in average of both seasons respectively, while GH2 recorded the highest concentrations of both Mn and Cu with values of 96.12 and 0.6 mg kg$^{-1}$ on average of both seasons respectively. Conversely, the application of GW recorded the lowest level of enhancement among the other treatments with Fe, Mn, and Cu, while K treatment registered the lowest level of Cu.

## Sugar beet productivity

The sugar beet productivity including root yield, shoot yield, and extracted sugar percentage are illustrated in Table 5, while sugar yield and sugar losses to molasses (SLM) are showed in Fig. 1. The ANOVA analysis showed significant ($p < 0.01$) effects of different treatments on those traits. The application of GH2 registered the highest values in root yield, shoot yield, and sugar yield recording 31.3 t fed$^{-1}$, 8.38 t fed$^{-1}$, and 5.69 t fed$^{-1}$ in the first season, and 33.86 t fed$^{-1}$, 11.32 t fed$^{-1}$, and 6.71 t fed$^{-1}$ in the second season respectively. In contrast, GW1 recorded the lowest enhancement. The application of G affected significantly in boosting the extracted sugar in the first season with an increment of 31.67% more than the control, while GH2 increased the extracted sugar in the second season with an increment of 17.57% compared to the control. On the contrary, the GW1 registered the lowest improvement in extracted sugar in both seasons. The sugar purity index has no significant differences between treatments in both seasons, while the SLM was affected by the application of GH treatments with GH2 recording the highest value (2.90%) in the first season, and GH1 recording the highest value (2.69%) in the second season. On the other hand, the application of G has no effects on the SLM compared to control in the first season, and GS1 in the second season. The sucrose content was affected significantly

**Table 3** Effect of different K- sources and glauconite extracts on soil physicochemical properties after two successful seasons of sugar beet plants.

| Treatments | Season 1 | | | | | | | Season 2 | | | | | | |
|---|---|---|---|---|---|---|---|---|---|---|---|---|---|---|
| | pH | EC | OM | ESP | BD | IR | SPR | pH | EC | OM | ESP | BD | IR | SPR |
| CK | 8.62[a] | 4.76[a] | 1.16[d] | 13.44[a] | 1.45[a] | 0.72[b] | 380.00[a] | 8.62[a] | 4.74[a] | 1.18[d] | 12.10[a] | 1.44[a] | 0.73[c] | 370.00[a] |
| K | 8.54[a] | 3.75[c] | 2.03[b] | 11.68[b] | 1.39[a] | 0.79[b] | 280.00[c] | 8.52[a] | 3.75[c] | 2.04[b] | 10.80[d] | 1.38[a] | 0.81[bc] | 270.00[c] |
| G | 8.53[a] | 3.36[c] | 2.36[a] | 10.57[c] | 1.34[a] | 1.08[a] | 250.00[c] | 8.52[a] | 3.35[c] | 2.42[a] | 10.21[h] | 1.33[a] | 1.11[a] | 240.00[c] |
| GS1 | 8.49[a] | 3.73[c] | 1.39[c] | 11.97[b] | 1.43[a] | 0.81[b] | 350.00[ab] | 8.49[a] | 3.72[c] | 1.41[cd] | 10.76[e] | 1.43[a] | 0.84[bc] | 340.00[ab] |
| GS2 | 8.49[a] | 3.65[c] | 1.47[c] | 12.04[b] | 1.43[a] | 0.84[b] | 350.00[ab] | 8.49[a] | 3.64[c] | 1.48[c] | 12.04[b] | 1.42[a] | 0.86[b] | 340.00[ab] |
| GW1 | 8.61[a] | 4.69[ab] | 1.18[d] | 10.28[c] | 1.45[a] | 0.72[b] | 370.00[ab] | 8.62[a] | 4.68[ab] | 1.19[d] | 10.25[g] | 1.44[a] | 0.74[bc] | 370.00[a] |
| GW2 | 8.61[a] | 4.25[b] | 1.19[d] | 10.66[c] | 1.44[a] | 0.81[b] | 350.00[ab] | 8.59[a] | 4.22[b] | 1.21[d] | 10.64[f] | 1.44[a] | 0.85[bc] | 350.00[ab] |
| GH1 | 8.41[a] | 3.39[c] | 1.87[b] | 11.49[b] | 1.41[a] | 1.07[a] | 340.00[ab] | 8.39[a] | 3.37[c] | 1.94[b] | 11.46[c] | 1.41[a] | 1.08[a] | 330.00[ab] |
| GH2 | 8.25[a] | 3.33[c] | 1.94[b] | 10.18[c] | 1.39[a] | 1.17[a] | 330.00[b] | 8.23[a] | 3.31[c] | 1.97[b] | 10.16[i] | 1.38[a] | 1.18[a] | 320.00[b] |
| Significance | ns | ** | ** | ** | ns | ** | ** | ns | ** | ** | ** | ns | ** | ** |
| LSD (0.05) | 1.01 | 0.46 | 0.20 | 0.76 | 0.17 | 0.11 | 39.77 | 1.01 | 0.46 | 0.27 | 0.01 | 0.17 | 0.11 | 38.87 |

**Notes.**

EC, Electrical conductivity (dS m$^{-1}$); ESP, Exchangeable sodium percentage (%); CEC, Cations exchange capacity (Cmol kg$^{-1}$); OM, organic matter (%); BD, Bulk density (g cm$^{-3}$); SPR, Soil penetration resistance (N cm$^{-2}$); IR, infiltration rate (cm h$^{-1}$).

Means in the same columns with different letter are significantly different at (0.05) level.

** Significant difference at 0.01 probability levels.

ns, not significant.

El-Sharkawy et al. (2025), *PeerJ*, DOI 10.7717/peerj.19452

**Table 4 Effect of different K- sources and glauconite extracts on soil nutrient availability (mg kg$^{-1}$) after two successful seasons of sugar beet plants.**

| Treatments | Season 1 | | | | | | | Season 2 | | | | | | |
|---|---|---|---|---|---|---|---|---|---|---|---|---|---|---|
| | NH$_4$ | NO$_3$ | K | Fe | Mn | Zn | Cu | NH$_4$ | NO$_3$ | K | Fe | Mn | Zn | Cu |
| CK | 27.55$^d$ | 20.66$^d$ | 181.35$^e$ | 3.96$^e$ | 23.01$^e$ | 0.18$^{bc}$ | 0.19$^f$ | 34.44$^d$ | 17.22$^d$ | 262.32$^e$ | 3.75$^e$ | 23.13$^e$ | 0.18$^d$ | 0.19$^e$ |
| K | 32.44$^{cd}$ | 28.80$^c$ | 398.53$^{dc}$ | 4.66$^{de}$ | 35.34$^d$ | 0.18$^{bc}$ | 0.27$^e$ | 34.44$^d$ | 34.44$^b$ | 506.61$^{bc}$ | 4.56$^{de}$ | 35.34$^d$ | 0.19$^d$ | 0.27$^d$ |
| G | 44. 77$^b$ | 43.22$^a$ | 374.74$^{cd}$ | 5.50$^{bc}$ | 58.98$^c$ | 0.22$^b$ | 0.34$^d$ | 61.92$^b$ | 30.96$^c$ | 470.34$^{cd}$ | 5.47$^{bc}$ | 59.03$^c$ | 0.36$^c$ | 0.47$^c$ |
| GS1 | 41.66$^b$ | 20.66$^d$ | 340.93$^d$ | 4.52$^{de}$ | 83.40$^b$ | 0.72$^a$ | 0.48$^c$ | 51.66$^c$ | 17.23$^d$ | 435.24$^d$ | 4.48$^{de}$ | 83.48$^b$ | 0.77$^b$ | 0.54$^b$ |
| GS2 | 55.48$^a$ | 17.28$^d$ | 343.83$^d$ | 12.12$^a$ | 82.33$^b$ | 0.78$^a$ | 0.53$^{bc}$ | 56.12$^c$ | 17.44$^d$ | 435.24$^d$ | 12.07$^a$ | 82.48$^b$ | 0.86$^a$ | 0.61$^a$ |
| GW1 | 34.44$^c$ | 17.22$^d$ | 193.44$^e$ | 4.29$^{de}$ | 30.15$^{de}$ | 0.18$^{bc}$ | 0.50$^{bc}$ | 34.44$^d$ | 17.22$^d$ | 262.27$^e$ | 4.18$^{de}$ | 30.16$^{de}$ | 0.20$^d$ | 0.52$^{bc}$ |
| GW2 | 34.44$^c$ | 17.22$^d$ | 193.58$^e$ | 4.90$^{cd}$ | 30.80$^{de}$ | 0.12$^c$ | 0.51$^{bc}$ | 34.43$^d$ | 17.24$^d$ | 286.81$^e$ | 4.89$^{cd}$ | 30.80$^{de}$ | 0.14$^d$ | 0.52$^{bc}$ |
| GH1 | 34.44$^c$ | 43.05$^a$ | 434.42$^{ab}$ | 5.64$^{bc}$ | 95.89$^a$ | 0.77$^a$ | 0.56$^{ab}$ | 34.44$^d$ | 51.67$^a$ | 554.58$^{ab}$ | 5.44$^{bc}$ | 96.00$^a$ | 0.78$^b$ | 0.57$^{ab}$ |
| GH2 | 51.66$^a$ | 34.44$^b$ | 472.97$^a$ | 5.72$^b$ | 96.12$^a$ | 0.77$^a$ | 0.60$^a$ | 68.88$^a$ | 18.24$^b$ | 579.25$^a$ | 5.72$^b$ | 96.12$^a$ | 0.78$^b$ | 0.60$^a$ |
| Significance | ** | ** | ** | ** | ** | ** | ** | ** | ** | ** | ** | ** | ** | ** |
| LSD (0.05) | 4.81 | 3.41 | 40.54 | 0.73 | 7.82 | 0.006 | 0.05 | 5.63 | 3.22 | 51.78 | 0.06 | 7.83 | 0.72 | 0.06 |

**Notes.**

Means in the same columns with different letter are significantly different at (0.05) level.

[**]Significant difference at 0.01 probability levels.

($p < 0.01$) by treatments in the first season, while it had no effects in the second season. The sucrose content recorded the highest values with GH2, G, and GH1 in the first season with no significant difference, and GH2 in the second season, while GW1 registered the lowest records in both seasons.

Figure 2 represents the root and shoot content of both Fe and Mn. Compared to different amendments, the data indicated that GH1 and GH2 increased both shoots and root contents of both elements in both seasons. While the K treatments recorded the lowest contents of both Fe and Mn in shoots in both seasons. The application of GW1 and GS1 recorded the lowest contents of Fe and Mn in roots with average increments of 13.06% and 3.8%, respectively, in both seasons.

## Sugar quality

The sugar quality parameters including alkalinity and sugar impurites (K, $\alpha$-NH$_4$, and Na) are presented in Fig. 3. Regarding the sugar impurities, all treatments affected significantly ($p < 0.01$) in K, $\alpha$-NH$_4$, and Na contents. It was noteworthy that all treatments increased the K and $\alpha$-NH$_4$ impurities compared to control with GH2 increasing both K, $\alpha$-NH$_4$ in both seasons recording average increments of 44.64% and 123.23% in the first and second seasons respectively. Regardless, the GW1 recorded the lowest levels of both K, $\alpha$-NH$_4$, but it was recorded the highest levels of Na with values of 5.41 and 3.55 Meq 100 g$^{-1}$ beet in the first and second seasons respectively. As for the alkalinity of sugar, the application of different treatments resulted in reducing the values compared to control with GH1 listed the lowest record (2.79) in the first season, and G (3.05) in the second season. The alkalinity of extracted sugar followed the same pattern in both seasons with GH2 recording the lowest values (2.79 and 3.41) and GW1 recorded the highest addition treatment with values 6.26 and 6.61 in the first and second seasons, respectively.

## DISCUSSION

Considering that foliar application is primarily designed for direct plant uptake, it can still indirectly impact soil properties, especially with substances like humic acid, sulfur, and glauconite extracts. While foliar application mainly targets plant tissues, excess nutrients or compounds applied to the leaves can leach or wash off into the soil, influencing its physical and chemical characteristics. The findings showed that different potassium (K) sources and glauconite extracts did not significantly affect soil pH over both seasons. This is consistent with previous studies indicating that soil pH is generally buffered and may remain stable with K or glauconite treatments due to their neutral to slightly alkaline properties (*Yang et al., 2020*; *Leon-Chang et al., 2022*). The organic matter (OM) was significantly enhanced by the application of glauconite, with values doubling compared to the control, as glauconite is known for its slow release of nutrients and its impact in improving soil structure (*Rudmin et al., 2019*), which also might explain its minimal impact on pH levels. The data also revealed that EC and ESP were significantly affected by the treatments ($p < 0.01$), highlighting their role in mitigating soil salinity. The foliar application with GH2 resulted in significant reductions of salinity in both seasons which could be explained by the role of humic substances as a chelating agent that can bind with sodium ions and other salts,

**Table 5  Sugar beet yield and productivity parameters as affected by K- sources and glauconite extracts after two successful seasons.**

| Treatments | 1st season | | | | | 2nd season | | | | |
|---|---|---|---|---|---|---|---|---|---|---|
| | Root yield (t fed$^{-1}$) | Shoot yield (t fed$^{-1}$) | Extracted sugar (%) | Sucrose (%) | Purity index (%) | Root yield (t fed$^{-1}$) | Shoot yield (t fed$^{-1}$) | Extracted sugar (%) | Sucrose (%) | Purity index (%) |
| CK | 15.61[d] | 4.30[d] | 11.40[b] | 14.42[c] | 86.86[a] | 17.42[h] | 5.82[h] | 14.32[b] | 17.12[b] | 90.26[a] |
| K | 23.80[b] | 6.38[b] | 13.81[a] | 16.93[ab] | 88.75[a] | 25.75[c] | 8.61[c] | 15.96[ab] | 19.03[ab] | 90.34[a] |
| G | 18.44[cd] | 4.94[cd] | 15.01[a] | 18.03[a] | 90.15[a] | 19.95[e] | 6.67[e] | 15.78[ab] | 18.97[ab] | 89.97[a] |
| GS1 | 17.55[d] | 4.70[d] | 13.63[a] | 16.75[ab] | 88.78[a] | 18.98[f] | 6.35[f] | 15.82[ab] | 18.66[ab] | 91.09[a] |
| GS2 | 20.43[c] | 5.47[c] | 13.75[a] | 16.94[ab] | 88.53[a] | 22.09[d] | 7.39[d] | 15.72[ab] | 18.68[b] | 90.65[a] |
| GW1 | 16.08[d] | 4.31[d] | 11.71[b] | 14.85[bc] | 86.73[a] | 17.42[h] | 5.82[h] | 14.33[b] | 17.20[ab] | 89.83[a] |
| GW2 | 16.15[d] | 4.33[d] | 11.75[b] | 14.94[bc] | 86.70[a] | 17.50[g] | 5.85[g] | 14.94[ab] | 17.90[ab] | 90.12[a] |
| GH1 | 24.15[b] | 6.59[b] | 14.26[a] | 17.59[a] | 88.64[a] | 26.13[b] | 8.74[b] | 16.13[ab] | 19.42[ab] | 89.62[a] |
| GH2 | 31.30[a] | 8.38[a] | 14.69[a] | 18.19[a] | 88.15[a] | 33.86[a] | 11.32[a] | 16.55[a] | 19.82[a] | 89.94[a] |
| Significance | ** | ** | ** | ** | ns | ** | ** | ns | ns | ns |
| LSD (0.05) | 2.48 | 0.67 | 1.59 | 1.96 | 10.44 | 0.06 | 0.01 | 1.83 | 2.19 | 10.69 |

Notes.

SLM, sugar losses to molasses.

means in the same columns with different letter are significantly different at (0.05) level.

[**]Significant difference at 0.01 probability levels.

ns, not significant.

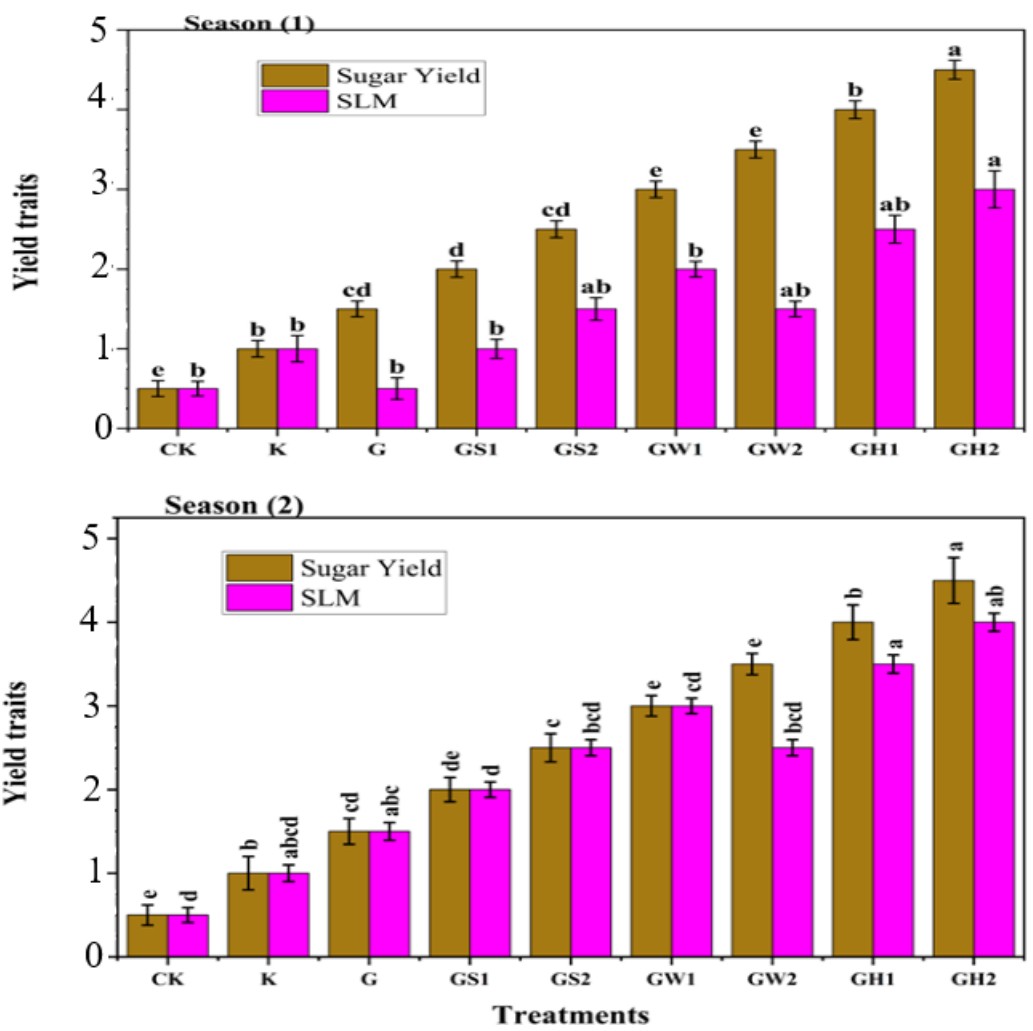

**Figure 1  The sugar yield (t fed −1) and sugar losses to molasses (SLM%) as affected by different K-sources and glauconite extracts after two successful seasons of sugar beet plants.**

making them less mobile and more likely to be leached out of the soil profile (*Huang, 2022*). Additionally, humic acid improves soil structure, increasing its porosity and permeability, which is evidenced by the enhancement occurred in water infiltration rate (IR) and facilitates the leaching of excess salts (*Sarlaki et al., 2024*) which in turn resulted in the enhancement of soil ESP. When humic acid enters the soil, it stimulates the formation of soil aggregates, which improve pore space for water movement (*Yang, Tang & Antonietti, 2021*). In contrast, the SPR showed fluctuations, with the lowest resistance recorded in the G and K treatments. This reduction in SPR suggests an improvement in soil fragmentation and reduction of soil compaction confirmed by the improvement configured in IR in both seasons, which is consistent with the findings of *Rudmin, Banerjee & Makarov (2020)* and *Chen, Banin & Borochovitch (1983)*, who reported that the application of potassium sources can enhance soil structure and reduce compaction. The hot water extraction

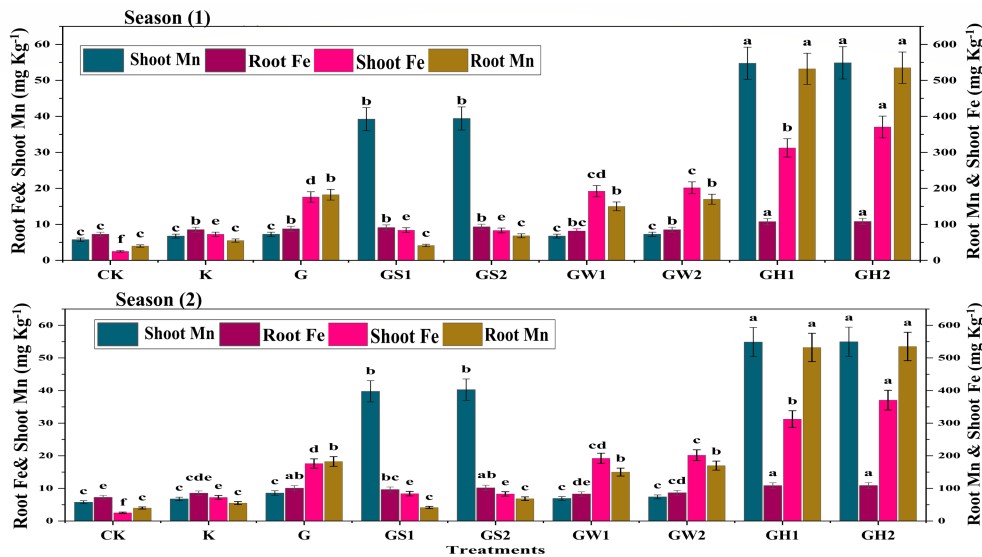

**Figure 2  The sugar quality characteristics as affected by different K- sources and glauconite extracts after two successful seasons of sugar beet plants.**

method may inadequately release structural components of glauconite, resulting in limited improvements to soil physical properties (*Hamed & Abdelhafez, 2020*), which reduces the chemical capability to significantly alter glauconite's matrix, thereby restricting its potential to enhance soil structure and porosity.

The observed variations in soil nutrient content across treatments are indicative of the unique contributions of glauconite, potassium sulfates, and glauconite extracts to nutrient availability and soil fertility. The divergences in nutrient availability can be attributed to the unique chemical characteristics of the extractors used, particularly the acidic nature of sulfuric and humic acids, which influence nutrient solubility and mobility in the soil. Foliar applications, though intended for plant uptake, often result in nutrient deposition onto soil surfaces, finding their way to the soil profile through irrigation, and influencing soil nutrient dynamics, especially when high-nutrient extracts like glauconite are used. The foliar applications with glauconite, especially in treatments GS2, G and GH2, demonstrated a significant influence on soil $NH_4^+$ and $NO_3^-$ levels in both seasons. The poor performance of GW treatments could be attributed to the limited efficacy of hot water extraction in releasing potassium and other nutrients from glauconite, which primarily relies on physical solubilization (*Rudmin et al., 2019*). This may result in insufficient release of key nutrients, such as potassium, reducing its effectiveness in improving soil fertility. This could be explained by the ability of sulfuric acid to break down soil organic matter partially and mineral structures in glauconite, thereby enhancing the release or replacement of bound nitrogen forms (*Hassan & Baioumy, 2006*). As well as the integration impacts between humic acid and glauconite facilitates the efficiency of glauconite-derived nutrients through the binding forces with elements (*Costa Júnior et al., 2024*) and increase the microbial population with its degradation capabilities of soil organic matter resulting in boosting

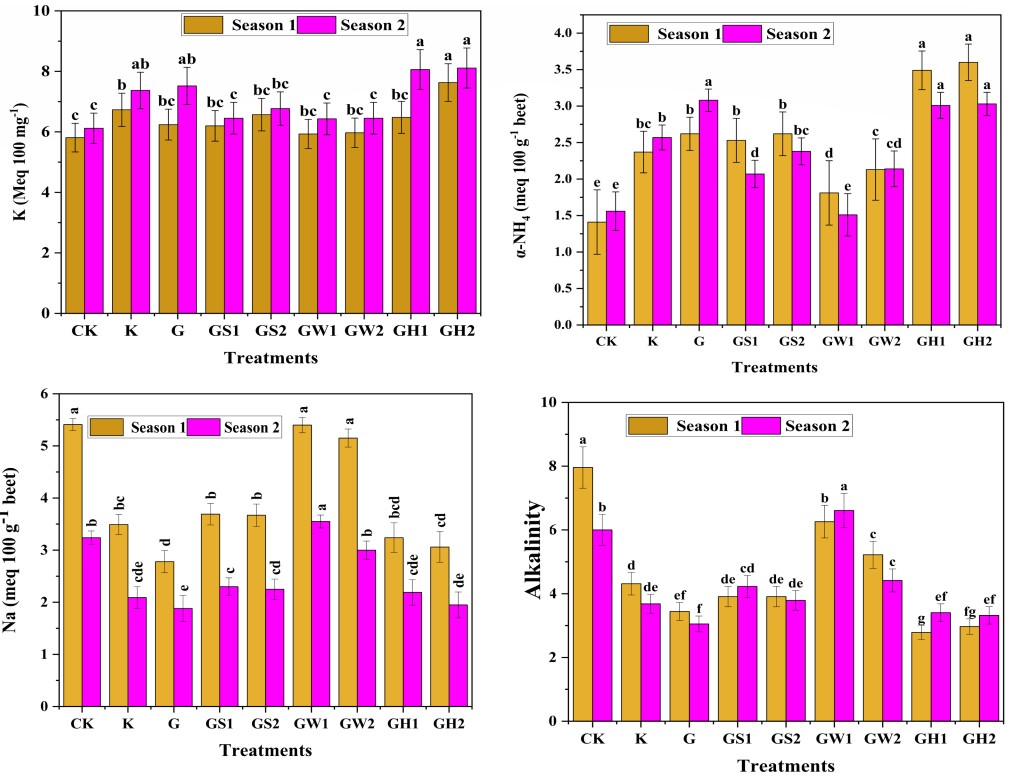

**Figure 3** The sugar beet micronutrient contents as affected by different K- sources and glauconite extracts after two successful seasons of sugar beet plants.

the soil nitrogen dynamics in both $NH_4^+$ and $NO_3^-$ forms. Potassium levels exhibited substantial increases in response to GH2, possibly due to the ability of humic acid to enhance the availability of potassium by chelating $K^+$ ions as presented in the GH compositions in Table 3 exhibiting the highest K content of 95 Meq $L^{-1}$ producing potassium humates, preventing them from leaching, and making them more accessible to plants (*Ampong, Thilakaranthna & Gorim, 2022*). Humic acids increase the cation exchange capacity (CEC) of soils, thus holding more $K^+$ in an available form (*Ali & Mindari, 2016*). This mechanism also explains why humic acid-extracted glauconite showed better performance in enhancing soil $K^+$ levels compared to water-extracted glauconite (GW1), which had the lowest $K^+$ enhancement. The hot water extraction method is less aggressive in breaking down the mineral matrix of glauconite recording the lowest K level in GW with value 4.8 Meq $L^{-1}$, limiting the release of potassium.

The results regarding micronutrient availability (Fe, Mn, Zn, and Cu) reveal significant variations depending on the type of extraction. The increase in Fe and Zn levels observed with GS2 treatment can be attributed to the strong leaching effects of sulfuric acid. Subsequently, sulfuric acid creates a highly acidic environment with more $H^+$ ions that are replaced in the reaction with a glauconite matrix with several elements including K, Fe, Zn, Mn, and Mg making them more available in the soil (*Costa Júnior et al., 2024*). This is particularly important for Fe, as its availability is highly pH-dependent

resulting from the high concentrations in GS composition, and at the same time, the acidification induced by sulfuric acid can solubilize iron oxides in the soil, thus increasing Fe concentrations (*Chen & Grassian, 2013*). In contrast, The GH composition recorded the highest Mn and Cu than other extraction methods pointing to the role of humic acids in mobilizing these micronutrients. Humic acids are known to form stable complexes with metal ions, particularly Mn and Cu, and prevent their precipitation or fixation in the soil, thus enhancing their bioavailability (*Dinu & Shkinev, 2020*). These chelating properties of humic acids not only increase the solubility of Mn and Cu but also protect them from being adsorbed onto soil particles, which explains the observed increases in their concentrations with GH2 applications. This is consistent with findings that humic substances can significantly influence the mobility and availability of trace metals in soils (*Hattab et al., 2014*). The overall lower performance of hot water (GW) extraction in enhancing nutrient availability compared to GS and GH treatments can be attributed to the less effective nature of water as an extractant. Water extraction primarily releases physically adsorbed ions without significantly breaking down the glauconite structure or chelating nutrients, which limits its ability to enhance soil nutrient levels. This is evident in the lower levels of Fe, Mn, and Cu observed in GW treatments, as compared to GS and GH treatments.

As for sugar beet (*Beta vulgaris* L.) productivity and sugar quality, the results demonstrate the positive impact of GH2 on yield components and extracted sugar, while indicating varied effects on sugar losses to molasses (SLM) and sucrose content across treatments. The increased root, shoot, and sugar yields observed with the GH2 treatment can be attributed to the role of humic acid as an organic enhancer, which improves nutrient availability, water retention, and overall soil health (*Ampong, Thilakaranthna & Gorim, 2022*), and the glauconite as inorganic enhancer rich in potassium rates with slow releasing functions. Both humic acid and potassium contents increase root development and nutrient uptake efficiency by enhancing the cation exchange capacity (CEC) and stimulating microbial activity, which aids in the release of nutrients that are essential for growth (*Selladurai & Purakayastha, 2016*). Enhanced root systems, in turn, allow the plants to access nutrients more effectively, thereby promoting greater shoot and sugar yields. Besides the role of both humic acid and K in sustaining the environmental stresses such as salinity. Conversely, the poor performance of GW treatments in enhancing plant productivity, particularly shoot and root yields, may be linked to insufficient nutrient release. Limited nutrient availability directly impacts plant growth, as key nutrients like K and Fe are critical for metabolic processes such as photosynthesis and enzyme activation (*Pandey, 2018*). The improvement in extracted sugar percentages, particularly with the G treatment in the first season and GH2 in the second season, suggests that the mineral and organic composition of glauconite treatments plays a crucial role in sugar quality with long residual effects with GH treatments. Glauconite, being a natural source of potassium and other minerals, is known to improve sugar content in sugar beet and other crops by enhancing osmotic balance and carbohydrate transport within the plant (*Shabana et al., 2024*). The presence of potassium is vital for sugar translocation and is directly linked to sugar accumulation in roots, which may explain the notable increase in extracted sugar with the G treatment

in the first season (*Xie et al., 2022*). Although the GH treatments recorded the highest sucrose and extracted sugar content, it recorded the highest SLM percentage in both seasons. Humic substances are known to mobilize micronutrients, including Fe, Mn, and Zn, which may play a role in the biochemical pathways involved in sucrose metabolism and molasses formation (*Makhlouf, Khalil & Saudy, 2022*). However, the magnitude of GH2 and GH1 treatments to increase SLM is consistent with the higher level of impurities ($\alpha$-NH$_4$ and K) that occurred in both seasons. These impurities, which might be resulted from the improvement of nutrient availability and uptake facilitated by the treatments, likely contributed to an increase in non-sucrose impurities. Such impurities can interfere with sucrose crystallization and increase molasses formation, thereby elevating SLM. This relationship aligns with previous findings that highlight the role of $\alpha$-amino nitrogen ($\alpha$-NH4), potassium (K), and sodium (Na) as key contributors to molasses formation due to their ability to disrupt sucrose recovery (*Muir, 2022*; *Shabana et al., 2024*; *Zhao et al., 2024*). While GH treatments recorded the lowest alkalinity levels in both seasons indicating a better magnitude for achieving high-quality sugar that meets industry standards (*Jaśkiewicz et al., 2024*). The ideal alkalinity level for high-quality sugar extraction is generally below 4.5 Meq/L (*Henke, Hinkova & Gillarova, 2019*). Lower alkalinity ensures that a greater proportion of sucrose crystallizes effectively and enhances the extractable sugar yield, while High alkalinity levels indicate the presence of non-sugar impurities, which can interfere with sugar crystallization and reduce overall sugar quality (*Velásquez et al., 2019*). Another factor that should be considered in the sucrose content with a significant increase in the first season but no effects in the second, is the seasonal variations. The enhanced sucrose content observed with GH2, G, and GH1 treatments indicates that humic acids and potassium-rich treatments can improve sugar accumulation by facilitating carbohydrate metabolism and sucrose translocation in sugar beet roots (*Shabana et al., 2024*). The lowest sugar beet yield and sucrose content observed with GW treatment, beside the remarkable high sodium impurity levels in extracted sugar, could be explained as the low nutritional content in this extraction method. Consequently, the GW-treated plants receive fewer nutrients, particularly potassium, which is crucial for plant productivity. The potassium deficiency may limit the ability of sugar beet to accumulate and transport sugars, reducing sucrose content and yield. In this circumstances if low potassium availability, plants often absorb more sodium to maintain osmotic balance and cellular functions (*Xie et al., 2022*), leading to an increase in sodium impurities in the roots, which detracts from sugar quality.

The heatmap presented in Fig. 4 reveals the important relationships between soil physico-chemical properties and sugar beet yield characteristics under various potassium sources and glauconite extracts over two seasons. A strong negative correlation of soil pH with root yield (−0.9109), sugar yield (−0.9013), and shoot yield (−0.7962), while EC showed a negative relationship with sugar yield (−0.721) and root yield (−0.6837). This suggests that slightly acidic to neutral levels improve sugar beet productivity by enhancing soil nutrient solubility and uptake (*Draycott & Christenson, 2003*; *Zi et al., 2024*). The adverse impact of EC emphasizes the need for treatments that reduce salinity levels, such as humic acid amendments, which help in maintaining nutrient balance and supporting healthier root development. Similarly, weak but consistent negative correlations with yield

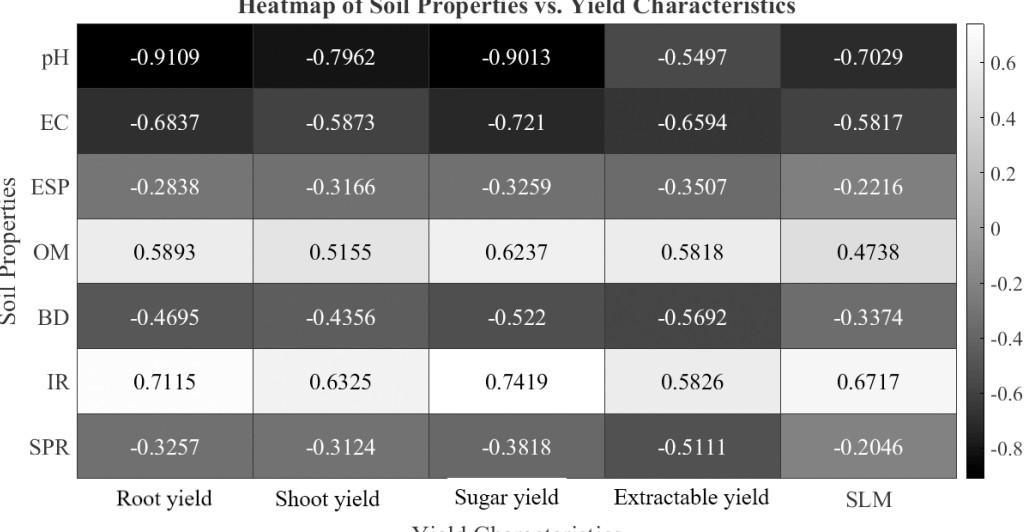

**Figure 4** Heatmap of the correlation between soil physico-chemical properties and sugar beet yield characteristics as affected by different K-sources and glauconite extracts after two successful seasons of sugar beet plants.

occurred with ESP, BD, and SPR. High BD indicates compacted soils, which restrict root expansion linked with poor soil structure and increasing soil SPR and ESP (*Vasu, Tiwary & Chandran, 2024*). On the contrary, OM and IR explored strong correlations with sugar beet yield particularly with root yield and sugar yield. The presence of organic matter improves soil structure, nutrient retention, and water-holding capacity, creating favorable conditions for improving the infiltration rate (IF) in soil and consequently enhancing plat yield (*Bashir et al., 2021*). This positive relationship highlights the role of soil conditioners, such as humic acid and glauconite, in enhancing soil permeability.

To delve into more outstandings of the correlations between more soil parameters and sugar beet productivity and sugar quality parameters, the use of a self-organizing map (SOM) in Fig. 5 as a kind of unsupervised machine learning used to cluster the data based on the optimum similarity, providing an in-depth understanding of the measured variables concerning the input parameters (*Klaine et al., 2017*; *Zi et al., 2024*). The SOM analysis, in the upper left map, clustered the dataset into four distinct groups, with cluster (2) revealed the higher significance and correlations, indicating potential differences in treatment effects or soil conditions within each group. Higher OM levels were observed in clusters associated with humic acid (GH) treatments, which improved soil structure, water retention, and nutrient (*Fageria, 2012*). Higher OM levels in clusters indicate the positive impact of organic amendments, such as humic acid, on soil health and crop productivity. Increased OM is closely associated with improved soil fertility, which is essential for sustaining crop yields, particularly in degraded soils. Maps with lower electrical conductivity (EC), pH, and ESP values were linked to maps with higher root, shoot, and sugar yields, underscoring the role of soil amendments in mitigating salinity and enhancing soil conditions. This

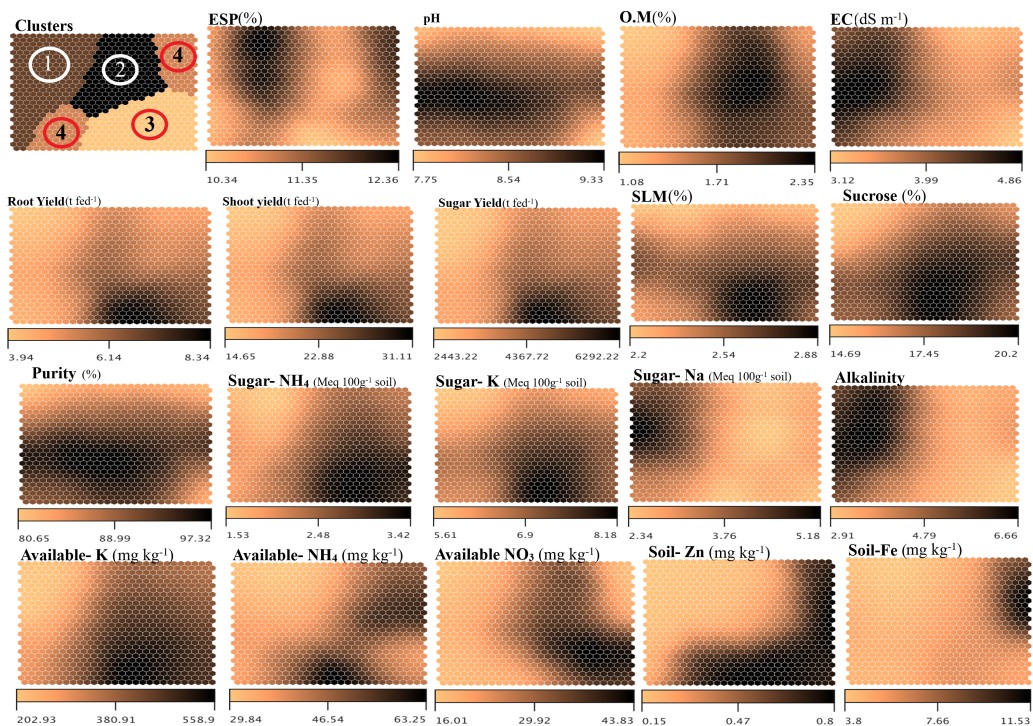

**Figure 5** Self-organizing map (SOM) of soil properties and both sugar beet productivity and sugar quality properties as affected by different K- sources and glauconite extracts after two successful seasons of sugar beet plants. ESP, exchangeable sodium percentage; OM, organic matter; EC, electrical conductivity; and SLM, sugar losses to molasses.

indicates that soil properties directly influence yield, with soil conditions favoring lower salinity, moderate pH, and higher organic content promoting better yields (*Nelson & Ham, 2000*). Conversely, the clusters' maps of soil rich in nutrients availability of K, $NH_4$, $NO_3$, Zn, and Fe are typically correlated with increased yield and quality in sugar beets, as these elements are crucial for various physiological functions, including photosynthesis, enzyme activation, and sugar metabolism (*Tariq et al., 2023*). Additionally, the maps emphasize that the reduction of SLM and increment of sucrose content and purity of extracted sugar are correlated with the reduction of sugar $NH_4$, sugar $NO_3$, Sugar Na, alkalinity, and the enhancement uptake of Fe, and Mn elements. This suggests that the treatments impact the uptake of macro and micro-nutrients which play critical roles in crop growth and sugar formation. These findings highlight the potential of organic and mineral amendments, such as humic acid and glauconite, to optimize soil conditions, mitigate salinity issues, and enhance nutrient availability, leading to higher and better-quality sugar beet yields.

The aforementioned relationships were specifically to describe the linkages between soil dynamics and sugar beet attributes. To deepen into the specific relationships between sugar beet plants conditions and the output sugar status, the non-metrical multidimensional scaling (NMDS) has been accomplished and presented in Fig. 6. The NMDS plots provide further insight into the relationships between the sugar beet plants as independent variables

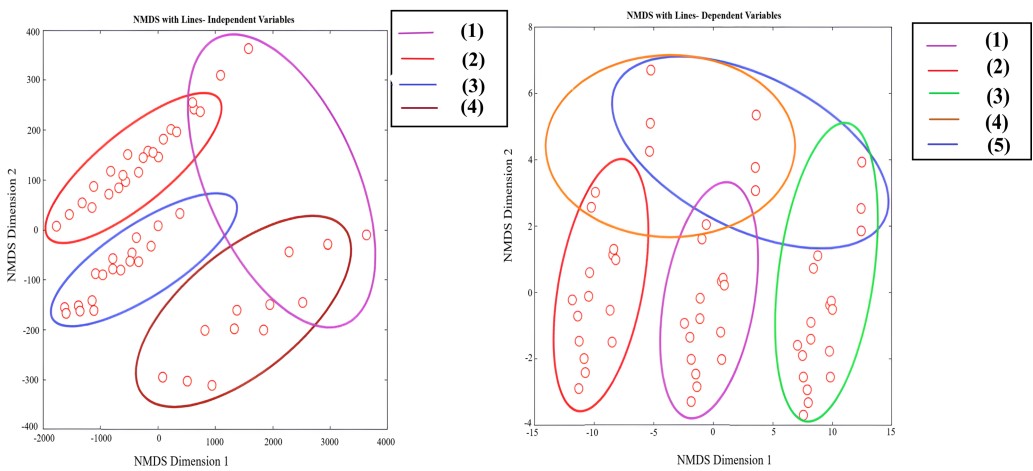

**Figure 6** Non-metrical multidimensional scaling (NMDS) of independent and dependent variables as affected by different K- sources and glauconite extracts after two successful seasons of sugar beet plants. For independent variables in four clusters including (1) Root yield and Shoot yield, (2) shoot yield and Shoot Mn, (3) shoot Fe, root Fe, and root Mn, and (4) sugar yield and root yield. The dependent variables in five clusters including (1) sugar $NH_4$ and sugar K, (2) sugar Na and alkalinity (3) % extractable sugar, (4) SLM, sucrose, (5) purity, and sugar K.

(*e.g.*, shoot yield, root yield, sugar yield, shoot Mn, shoot Fe, root Fe, and root Mn) and the extracted sugar as dependent variables (*e.g.*, % extractable sugar, SLM, sucrose, purity, sugar K, sugar Na, sugar $NH_4$, and alkalinity) under the influence of different potassium sources and glauconite extracts over two seasons of sugar beet cultivation. The Figure 6A shows distinct groupings with overlapping clusters, indicating that certain treatments have a pronounced effect on variables such as shoot and root yield, along with micronutrient accumulation in plant tissues. The clustering patterns here imply that specific potassium sources and glauconite treatments are enhancing the uptake of Mn and Fe in both shoot and root tissues, intunrs influencing plant growth and yield (*El-Mageed et al., 2022*). Notably, the grouping of root yield and shoot yield in proximity within specific clusters (as shown in purple groups) suggests that treatments might enhance both biomass and nutrient allocation in sugar beet plants. Additionally, Fig. 6B further emphasizes the effect of these treatments on sugar-related attributes such as extractable sugar, SLM, and sucrose. Clusters with high sucrose and extractable sugar content (shown in green grouping line) reflect optimal growing conditions where nutrient availability and soil amendments, like glauconite extracts, positively affect sugar quality metrics. Treatments that enhance potassium (K) availability are crucial since potassium regulates osmotic balance and sugar translocation within the plant, which are critical for increasing both sucrose content and purity (*Kaur et al., 2021*). Additionally, the clustering patterns in variables like sugar Na and alkalinity (shown in red grouping line) provide insights into how different treatments impact the ion balance, which directly affects SLM and the overall extractable sugar yield. The presence of distinct clusters for sugar $NH_4$ andsugar K (shown in purple grouping line) suggests that certain treatments are particularly effective in influencing these variables.

For example, enhanced levels of potassium in sugar tissues are linked to increased sugar quality and lower sugar losses, indicating that specific K-sources contribute to better ion regulation within the plant (*Shabana et al., 2024*). These findings underscore the potential benefits of combining K-sources with glauconite to enhance both the yield and quality of sugar beet crops, especially in challenging soil environments.

## CONCLUSIONS

A two seasons field experiment was conducted to investigate the effects of different potassium (K) sources and glauconite extracts on sugar beet yield and quality grown in saline soils. Potassium sources include traditional potassium sulfate (K), glauconite (G) and different glauconite extracted using sulfuric acid (GS), humic acid (GH), or hot water (GH) in two foliar concentrations (20- and 40-mL $L^{-1}$). The findings revealed that GH and G treatments improved soil properties by reducing the EC, ESP, and BD, while increased OM and IR levels. The findings investigated that despite glauconite rock is a cheap waste, when extracted with humic acid, a highly concentrated extract of trace elements is obtained, which could be used as a foliar fertilizer alternatively to manufactured trace element fertilizers, as they are more concentrated, easier to absorb, and less expensive compared to the mineral elements available in the markets. The GH2 also improved sugar beet yield attributes recording average enhancements of root yield (94.84%), shoot yield (100.45%), and total sugar yield (137.22%) compared to control, while GW recorded the lowest improvement. Additionally, although the GH2 recorded the highest SLM with highest sugar impurities of K and $\alpha$-NH$_4$, it recorded the lowest alkalinity levels which is preferable for sugar industries. The NMDS plots emphasized the high impacts of different treatments particularly GH treatments in both sugar beet plants productivity, sugar yield and sugar quality progress. These findings underscore the potential for integrating natural resources like glauconite with eco-friendly extractants such as humic acid into sustainable agricultural practices. Future research should focus on long-term studies across various agro-ecological zones to validate these results under different climatic and soil conditions. Investigations into alternative eco-friendly extraction methods, such as microbial or enzymatic processes, are necessary to enhance glauconite's nutrient availability while minimizing environmental impacts.

### Funding
This work was supported by Princess Nourah bint Abdulrahman University Researchers Supporting Project number (PNURSP2025R101), Princess Nourah bint Abdulrahman University, Riyadh, Saudi Arabia. The funders had no role in study design, data collection and analysis, decision to publish, or preparation of the manuscript.

### Grant Disclosures
The following grant information was disclosed by the authors:

Princess Nourah bint Abdulrahman University Researchers Supporting Project: PNURSP2025R101.
Princess Nourah bint Abdulrahman University, Riyadh, Saudi Arabia.

## Competing Interests

The authors declare there are no competing interests.

## Author Contributions

- Mahmoud El-Sharkawy conceived and designed the experiments, analyzed the data, prepared figures and/or tables, authored or reviewed drafts of the article, funding acquisition, Supervision, and approved the final draft.
- Modhi O. Alotaibi conceived and designed the experiments, analyzed the data, prepared figures and/or tables, funding acquisition, and approved the final draft.
- Esawy Mahmoud conceived and designed the experiments, analyzed the data, prepared figures and/or tables, authored or reviewed drafts of the article, and approved the final draft.
- Kholoud A. El-Naqma performed the experiments, authored or reviewed drafts of the article, and approved the final draft.
- Ramadan E. Kanany conceived and designed the experiments, performed the experiments, authored or reviewed drafts of the article, and approved the final draft.
- Medhat G. Zoghdan performed the experiments, authored or reviewed drafts of the article, and approved the final draft.
- Mahmoud M. Shabana conceived and designed the experiments, performed the experiments, authored or reviewed drafts of the article, and approved the final draft.

## Data Availability

The raw data for the experimental traits and variables for both soil and sugar beet plants are available in the Supplementary Files.

## Supplemental Information

Supplemental information for this article can be found online at http://dx.doi.org/10.7717/peerj.19452#supplemental-information.

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
