# Peer review of "Novel glauconite compounds improve soil properties and sugar beet (Beta vulgaris L.) yields in saline soils"

_PeerJ, doi:10.7717/peerj.19452_

## Round 0.1 · original submission · Major Revisions

Dear Dr. El-Sharkawy

It is my opinion as the Academic Editor for your article - Impact of novel glauconite extraction models on soil properties and sugar beet (Beta vulgaris L.) yields in saline soils - that it requires several major and minor changes. The reviewers' have pointed out a range of shortcomings in your manuscript that need to be properly addressed before it is accepted for publication.

You are therefore requested to go though these suggestions and modify your manuscript, taking into account all the required changes. You need to place equal emphasis on improving each and every section of the article with focus on those relating to experimental methods and procedures.

In addition to the reviewers' comments, I have following suggestions for your consideration:

The manuscript title 'Impact of novel glauconite extraction models on soil properties and sugar beet (Beta vulgaris L.) yields in saline soils' may be changed to 'Novel glauconite compounds improve soil properties and sugar beet (Beta vulgaris L.) yields in saline soils'. Moreover, you should elaborate the statistical analysis part in order to make it more comprehensive.

You are also informed that your revised submission will undergo additional peer reviews in order to ensure that the modifications you have made are up to the mark.

Reviewer 1 ·

Basic reporting

The manuscript presents important findings on the use of glauconite extracts to improve soil health and sugar beet productivity in saline soils. However, improvements are needed in language clarity, methodological details, and the discussion of practical implications. Addressing these points will enhance the manuscript's impact and readability.
The introduction provides sufficient background but could benefit from a more focused articulation of the knowledge gap and how this research fills it. Suggest extending lines 78–83 to clarify the novelty of the glauconite extraction techniques.

• Grammar and Language Issues:
o In the abstract (lines 32–33), "GH2 enhanced soil nutrient availability including N (73.4%)...". Consider rephrasing as "GH2 treatment improved soil nutrient availability, notably increasing nitrogen (by 73.4%)..."
o Line 71: "The nutrients availabilities in solid form of glauconite might be limited due to its low solubility." This should read, "The nutrient availability in the solid form of glauconite may be limited due to its low solubility."
o Line 345: "The magnitude of GH2 and GH1 treatments to increase SLM is in convenient with..." The term "in convenient" is awkward; replace it with "consistent with."

Experimental design

• The authors claim that "various extraction techniques...could enhance its impact on plant growth" (line 72). However, the manuscript does not justify the choice of extraction methods (sulfuric acid, humic acid, and hot water). Include references or reasoning for selecting these methods.

Validity of the findings

• Some claims lack sufficient support. For example, line 345 states that impurities increased SLM, but this relationship needs more explanation or references.
• The manuscript discusses the positive impacts of GH2 extensively but does not adequately address why GW treatments performed poorly. Add a discussion on potential limitations of the hot water extraction method.
• Conclusions drawn from NMDS and SOM analyses (lines 384–436) are difficult to follow without clearer visuals or summaries. Simplify or expand explanations for these results.

Additional comments

o Some figures (e.g., Figure 1) are difficult to interpret due to small font sizes. Consider improving visual clarity.

Reviewer 2 ·

Basic reporting

I have carefully reviewed the manuscript entitled “Impact of novel glauconite extraction models on soil properties and sugar beet (Beta vulgaris L.) yields in saline soil.” The manuscript has the potential to get published, however it needs some major corrections before its publication. It is also advised for authors to thoroughly go through the MS and correct the spelling as well as grammatical errors. The decision over the MS is the major revision.

Experimental design

In the introduction part, the authors should provide a declaration of uniqueness of the present study.
Write the Novelty of the current study.
Write a paragraph about the previous report along with the research gap with the significance of the current research.
The keywords are missing.
In materials and methods section, all the procedures should be supported by the references.
Write the complete instrument details wherever necessary.
Why flame photometer was used and write the instrument details.

Validity of the findings

Line no. 181-183, explain in detail.
Explain the line no. 197-200 and also correct the spellings wherever necessary.
Correct the spellings of “increase” and “value” in line no. 202 and 203.
In line no. 213, it should be 6.71.
Explain the line no. 214 -216.
Recommend the specific future research direction in the conclusion part.
The English of the MS should be corrected by a native English speaker.
The references need to go thoroughly again, as some are not in the journal format.

Additional comments

NA

---

## Round 0.2 · Minor Revisions

Dear Dr. El-Sharkawy,

Thank you for your submission to PeerJ.

It is my opinion as the Academic Editor for your article - Novel glauconite compounds improve soil properties and sugar beet (Beta vulgaris L.) yields in saline soils - that it requires a few Minor Revisions.

Therefore, you are advised to carry out the revision keeping reviewers' comments in mind.

Reviewer 1 ·

Basic reporting

I have reviewed the revised version of the manuscript and confirm that the authors have adequately addressed all my comments and suggestions. The revisions improve the clarity and quality of the paper, and I have no further concerns at this stage.

Experimental design

.

Validity of the findings

.

Additional comments

.

Reviewer 2 ·

Basic reporting

Dear Author,
I have carefully reviewed the MS entitled “Novel glauconite compounds improve soil properties and sugar beet (Beta vulgaris L.) yields in saline soils”. The MS was found to be suitable for publication after the correction of the minor mistakes. However there are minor mistakes in the MS, which the author need to rectify in the journal as per the comment given below:

1. Krupskaya et al., 2018; Havre, (1961); Hamed & Abdelhafez, 2020; Pandey, 2018; references are missing in the reference section.
2. In line no. 261, it should not be in capital.
3. In line no 450-452, why some of the words are in capital and some in small. Kindly maintain the uniformity.
4. Either add the reference “Page 1982” in the MS or remove from the reference section. As it is not present in the MS.

Experimental design

NA

Validity of the findings

NA

Additional comments

NA

---

## Round 0.3 · Minor Revisions

Dear Dr. El-Sharkawy,

Thank you for your submission to PeerJ.

I am writing to inform you that your manuscript - Novel glauconite compounds improve soil properties and sugar beet (Beta vulgaris L.) yields in saline soils - is almost ready to be Accepted for publication.

Before I do so, the Section Editor, has commented and said:

"This is an interesting paper that needs a few minor modifications before publication.

+++ 1 The abstract defines the GH treatment but then details results for GH2 without defining what the "2" means.

+++ Figure 3 (heat map). Please indicate in the legend that the numbers represent correlations. Also, the color scale needs to be changed. Having red at the top and a reddish color at the bottom is confusing. Plus, there are too many colors. i would recommend a 2 color gradient with one color for the negative values and another color for the positive values (fading to either white or black for values near 0). Avoid red/green because of red/green colorblindness. magenta/green or red/blue would work.

+++ Figure 4 legend: what does each "cell" in the SOM represent? Please explain in legend

+++ Figure 5 and legend. Please indicate in the legend what the colors mean. Also, how do the clusters relate to treatment? Coloring the points by treatment would be informative.

+++ Although not a requirement, it would be really nice to see some of the yield traits (Table 5) presented in a figure."

Please address these items in a minor revision.

Reviewer 2 ·

Basic reporting

Dear Authors,
I have carefully reviewed the MS entitled “Novel glauconite compounds improve soil properties and sugar beet (Beta vulgaris L.) yields in saline soils”. I have thoroughly reviewed the manuscript and found that all the requested changes have been correctly implemented. Therefore, the manuscript can be ready for publication.

Experimental design

NA

Validity of the findings

NA

Additional comments

NA

---

## Round 0.4 · Minor Revisions

Dear Dr. El-Sharkawy,

Thank you for your submission to PeerJ.

We notice that the "new" heat map (now figure 4) is the same as the old heatmap. Perhaps the authors mistakenly uploaded the old version.

---

## Round 0.5 · accepted · Accept

Dear Dr. El-Sharkawy,

Thank you for your submission to PeerJ.

I am writing to inform you that your manuscript - Novel glauconite compounds improve soil properties and sugar beet (Beta vulgaris L.) yields in saline soils - has been Accepted for publication.

Congratulations!